# Sky and Ground Segmentation in the Navigation Visions of the Planetary Rovers

**DOI:** 10.3390/s21216996

**Published:** 2021-10-21

**Authors:** Boyu Kuang, Zeeshan A. Rana, Yifan Zhao

**Affiliations:** 1Centre for Computational Engineering Sciences (CES), School of Aerospace, Transport and Manufacturing (SATM), Cranfield University, Bedfordshire MK43 0AL, UK; zeeshan.rana@cranfield.ac.uk; 2Centre for Life-Cycle Engineering and Management, School of Aerospace, Transport and Manufacturing (SATM), Cranfield University, Bedfordshire MK43 0AL, UK; yifan.zhao@cranfield.ac.uk

**Keywords:** semantic segmentation, weak supervision, transfer learning, conservative annotation method, visual navigation, visual sensor

## Abstract

Sky and ground are two essential semantic components in computer vision, robotics, and remote sensing. The sky and ground segmentation has become increasingly popular. This research proposes a sky and ground segmentation framework for the rover navigation visions by adopting weak supervision and transfer learning technologies. A new sky and ground segmentation neural network (network in U-shaped network (NI-U-Net)) and a conservative annotation method have been proposed. The pre-trained process achieves the best results on a popular open benchmark (the Skyfinder dataset) by evaluating seven metrics compared to the state-of-the-art. These seven metrics achieve 99.232%, 99.211%, 99.221%, 99.104%, 0.0077, 0.0427, and 98.223% on accuracy, precision, recall, dice score (F1), misclassification rate (MCR), root mean squared error (RMSE), and intersection over union (IoU), respectively. The conservative annotation method achieves superior performance with limited manual intervention. The NI-U-Net can operate with 40 frames per second (FPS) to maintain the real-time property. The proposed framework successfully fills the gap between the laboratory results (with rich idea data) and the practical application (in the wild). The achievement can provide essential semantic information (sky and ground) for the rover navigation vision.

## 1. Introduction

Sky and ground segmentation is a popular topic in computer vision that has a wide range of applications [1,2,3,4,5,6,7,8,9,10,11]. Sky and ground are the two essential components in outdoor and remote scenes [11,12,13,14,15,16]. Furthermore, sky and ground segmentation research is also an active topic in bionics studies [17,18,19,20]. Figure A1 displays six views from the Perseverance, Opportunity, and Spirit Mars rovers of the National Aeronautics and Space Administration (NASA). The visual environment of a planetary rover is complex [21,22]. Semantic segmentation helps planetary rovers understand the environment logically.

This research is inspired by the following related studies on sky or ground segmentation (Figure 1 depicts the summary of the related studies). Many studies focused on the support vector machine (SVM). McGee et al. [23] and Liu et al. [24] used SVM for sky segmentation. De Mattos et al. [25] used a whiteness-based SVM. Song et al. [26] used two imbalance SVM classifiers and similarity measurement. Ye et al. [6] and Dev et al. [27] focused on The whole-sky imager and proposed a superpixel-based SVM method. Beuren et al. [28] proposed a whiteness- and blueness-based SVM method. Deep learning technology also produces an increasingly significant role in environmental remote sensing as “big data” [29,30]. Tighe and Lazebnik [10] proposed a non-parametric method combined with Markov Random Field (MRF) to segment sky pixels simply and efficiently. Mihail et al. [31] proposed an open segmentation dataset with ground-truth annotations and proposed a benchmark by comparing three popular segmentation methods (this research uses the dataset and the benchmark). Tsai et al. [2] proposed a sky segmentation classifier based on fully convolutional networks (FCN) and conditional random field (CRF) models. Liu et al. [7] used image matting and region growth algorithms to achieve sky segmentation. Vargas-Munoz et al. [32] proposed a framework based on the iterative spanning forest (ISF). Dev et al. [12] proposed a color-based segmentation method. Fu et al. [33] used a method based on a gray threshold. Place et al. [34] proposed a RefineNet-based solution. Nice et al. [35] proposed a new sky pixel detection system that can select mean-shift segmentation, K-means clustering, and Sobel filters to detect sky pixels. Hożyń and Zalewski [36] proposed a solution of adaptive filtering and progressive segmentation.

The SVM has good performance in small datasets, but it becomes increasingly slow as the dataset increases [37]. SVM uses the kernel matrix of the dataset to describe the similarity between samples. Therefore, the number of matrix-elements increases with the square of the data size. Besides, many of the above methods (including SVM) require supports from feature engineering. Thus, the corresponding solution is not End-to-End (E2E). The E2E refers to the functionality of inputting raw ultrasonic signals then outputting the identified flow regime [38]. The feature engineering used in most research is artificial features, and these feature extraction methods require many manual adjustments. The feature engineering selectively retains and discards information, which leads to a high-level of information loss. Furthermore, the existing deep learning solutions require high-level feature engineering (such as color, superpixels, and grayscale), limiting the generality. Besides, the deep learning solutions highly rely on post-processing (such as CRF), which limited their online deployment.

Some related studies focus on the sky and ground segmentation in the planetary rover scenario. Shen and Wang [40] proposed a gradient information and energy function optimization method in a single image for sky and ground segmentation. Ahmad et al. [41] proposed a fusion of edge-less and edge-based approach to detect the horizon. Shang et al. [42] proposed a superpixel-based approach, which works for robotic navigation. However, these studies are not E2E solutions. Carrio et al. [43] proposed a method for sky segmentation for unmanned aerial vehicles (UAVs). Chiodini et al. [44] proposed a method to detect the horizon to support the localization for Mars rover. The practical scenarios for these studies are different from the planetary rover navigation vision. Verbickas and Whitehead [45] proposed a convolutional neural network (CNN)-based solution. However, this solution cannot guarantee the scale-invariance between the input image and output prediction. Moreover, the dataset applied in their research contained multiple scenes, the space scenario data had a limited amount. Thus, it is not easy to achieve a significant segmentation performance in planetary rover navigation vision.

Moreover, the related solutions are mostly in the laboratory environment, and the data in the practical scenes are not annotated. The above-related studies mainly use supervised learning, which requires a large amount of well-annotated data. Cordts et al. [46] claim that an image in the Cityscape dataset takes 1.5 h for annotation. Manual data annotation is inefficient, and human factors will reduce its reliability. There are many existing data for planetary rover navigation, but they have not been well annotated with pixel-level. Therefore, there is a significant gap between a large amount of raw data and well-annotations, which is also a major challenge occurring in many other engineering and industrial missions. This research aims to fill the gap between the lavatory achievement (open benchmark) and the practical application (navigation vision of the planetary rovers) through the proposed framework using the conservative annotation method, weak supervision, and transfer learning.

This research proposes a sky and ground segmentation framework based on weak supervision and conservative annotation for the navigation visions of the planetary rover, which can improve the accuracy and efficiency of semantic landmarks detection in the practical scenario. The contributions of this research mainly include the following points.

(1)This research appears to be the first successful E2E and real-time solution for the navigation visions of the planetary rover.(2)The sky and ground segmentation network has achieved the best results in seven metrics on the open benchmark (the Skyfinder dataset) compared to the state-of-the-art.(3)This research designs a new sky and ground segmentation neural network (network in U-shaped network (NI-U-Net)) for the sky and ground segmentation.(4)This research proposes a conservative annotation method correlated with the conservative cross-entropy ($loss_{con}$) and conservative accuracy ($acc_{con}$for the weak supervision. The results indicate that the conservative annotation method can achieve superior performance with limited manual intervention, and the annotation speed is about one to two minutes per image.(5)This research further transfers the pre-trained NI-U-Net into the navigation vision of the planetary rovers through transfer learning and weak supervision, forming an E2E sky and ground segmentation framework that meets real-time requirements.

The discussion is arranged as follows. Section 2 introduces the methods used in this research. Section 3 discusses the experimental results. The conclusion is drawn in Section 4. The appendix contains the pseudo-code for the conservative annotation algorithm and some figures. Video data is attached as electronic materials.

## 2. Methods

The proposed sky and ground segmentation framework for the navigation visions of planetary rover adopts two datasets, the Skyfinder dataset [31] and the Katwijk beach planetary rover dataset [47]. First, this research designs a sky and ground segmentation neural network (in Section 2.2) pre-trained using a large and annotated dataset (the Skyfinder dataset). Second, this research proposes a conservative annotation method for annotating sky and ground pixels in the practical navigation vision of the planetary rovers (in Section 2.3). It is noteworthy that this research utilizes the Katwijk beach planetary rover dataset as the practical planetary scene. Then, this research conducts the data augmentation. Finally, this research uses the augmented data to perform weak supervision on the pre-trained sky and ground segmentation network, which transfers the prior knowledge from the pre-training process into the practical navigation scenario. Figure 2 depicts the process of the proposed overall framework. Semantic segmentation tasks of unstructured scenes usually do not have pixel-level annotations. The Katwijk dataset is the representation of unstructured scenes in this research. Traditional supervised learning is difficult to be directly used in this type of task. Although the method based on multiple labelers can reduce the human error in the annotation, it is difficult to eliminate the error in some complex scenes, even with various labelers. On the other hand, the unstructured scenes share a similar prior knowledge of sky and ground segmentation as the Skyfinder dataset, and the Skyfinder dataset is well annotated. So, a solution with a weakly supervised architecture based on transfer learning becomes very promising.

### 2.1. Datasets

#### 2.1.1. The Skyfinder Dataset

The Skyfinder dataset is the largest existing pixel-annotated sky segmentation dataset (containing weather influences). The Skyfinder dataset contains approximately 90,000 images taken from 53 different scenes (in the wild). The ground-truth labels have been binarily annotated the sky and non-sky pixels. It is noteworthy that this research names the non-sky pixels as the ground pixels for better discussion. The Skyfinder dataset is a widely used dataset in sky and ground segmentation [5,26,31,34,35].

According to the authors of the Skyfinder dataset [31], this research eliminates some images because of file damage, black screen, overexposure, and significantly upset weather. Figure A2 displays the examples of eliminated images. The same action for eliminating the “bad” images was also conducted in the related studies [5,26,31,34,35], which is a routine for using the Skyfinder dataset. Besides, the upset images (Figure A2) do not stay in the scope of this research. This research selects nearly 70,000 images from 53 scenes, which is 10% to 50% more selected ratio than in the related studies [5,26,31,34,35]. Thus, the pre-training in this research is more challenged (see detailed results and discussion in Section 3.1). Figure 3 displays some difficult scenes in the Skyfinder dataset. Although the Skyfinder dataset captures images in the Earth, it shares a similar logic of sky and ground segmentation in the navigation vision of the planetary rovers.

This research divides the pre-trained dataset into the training set (about 50,000 images), the validation set (about 10,000 images), and the testing set (about 10,000 images) according to the ratio of 70%, 15%, and 15% (after a randomized shuffle). The testing set is independent of the training process to provide an independent evaluation.

Various images of an individual scene in the Skyfinder dataset share the same sky and ground annotation (see [31] for details). The distinction between different images of the same annotation comes from weather, illumination, moving targets, etc. Technically, the input of these distinguishes can be considered as some “data augmentations.” Thus, an overall shuffle might introduce the risk of data contamination. However, dividing the training, validation, and testing set according to various scenes does not make sense because it is not comparable among different scenes. This research eliminates this concern through an experiment using the U-Net, and the discussion can be found in Section 3.1.

#### 2.1.2. The Katwijk Beach Planetary Rover Dataset

The Katwijk beach planetary rover dataset is a widely used dataset for navigation vision of the planetary rovers [48,49,50,51,52,53]. The Katwijk dataset is a proper dataset as the practical condition of planetary exploration. The Katwijk dataset was captured on a section of beach near Katwijk, the Netherlands, where contains natural, unstructured, and sandy terrains. The European Space and Technology Research Center also developed their heavy-duty planetary rover (HDPR) platform near the Katwijk beach [47]. The visual content of the Katwijk dataset covers most of the terrains in the study of Ono et al. [54]. Hewitt et al. claimed that the Katwijk dataset can emulate a Mars-analog environment, and is more representative to the planetary rovers [47]. Lamarre et al. also use the Katwijk dataset as a reference when they propose their “the Canadian Planetary Emulation Terrain Energy-Aware Rover Navigation Dataset” [50]. Chiodini et al. and Furlán et al. use the Katwijk dataset as the planetary terrain in their studies [52,53].

Another two similar planetary rover datasets are [55] and [50], but they are not suitable for this research because they contain many irrelevant targets, such as people, plants, buildings, vehicles, stagnant water and snow. Figure A3a,b illustrate the example images of [50,55], respectively. The red arrows and numbers highlight some examples of irrelevant targets. Besides, the NASA image album [56] may provide many images from Mars or Moon, but they are individual images instead of the format of video stream. Thus, this dataset is not fit the aim of evaluating the integration and real-time property in a planetary rover-based platform.

The Katwijk dataset [47] contains three traverses, and its visual content is only composed of rocks and sands. Its cameras include a panoramic camera (PanCam) with a wide field of view and a localization camera (LocCam) for visual localization, and both of them are binocular cameras. This research selected camera No. 0 in the LocCam as the only source data, which has a total of about 22,000 images. Hewitt et al. [47] use Bumblebee2 stereo camera as the LocCam, and the resolution is 1024×768 pixels.

### 2.2. Pre-Training Process: Sky and Ground Segmentation Network

The proposed sky and ground segmentation network consists of multiple convolutional networks (ConvNets). The proposed sky and ground segmentation network has been inspired by the U-shaped network (U-Net) [57,58,59] and network in network (NIN) [60]. U-Net has been widely used in semantic segmentation applications [61,62]. Figure 4 depicts the structure of the proposed sky and ground segmentation network. Moreover, the proposed network does not directly use a ready-made ConvNets architecture.

The main inspirations from the U-net [57] are in two aspects. First, the entire network (see Figure 4) adopts the overall configuration of the encoder-decoder [63,64]. The encoder structure has good information mining capability, which compresses the scale in the length and width directions and expands the depth direction scale. This can greatly increase the receptive field of the encoder structure, thereby numerically realizing a large range of image information interaction (even if they are originally located in different image regions). Second, U-net constructs a “highway” from the lower-level convolution structure to the upper-level convolution structure through the concatenation (the purple arrows in Figure 4) between the encoder and decoder tensors with the same scale (width and height). This can ensure stronger gradient feedback, thereby avoiding the vanishing gradients caused by the deep architectures.

The inspirations from NIN [60] are in two aspects. First, this research locates the micro-networks (the network in network, NIN) followed by the scale-changing ConvNets (see the orange square in Figure 4 and Figure 5a). The level of abstraction is low in traditional CNN (with the typical convolutional and pooling design). [60] NIN structure can enhance the discriminability for local patches within the receptive field. Second, this research put two 1 × 1 ConvNets in all NINs. The 1 × 1 ConvNets is a cascaded cross channel parametric pooling on a normal convolution layer. This cascaded cross-channel parametric pooling structure allows complex and learnable interactions of cross-channel information. Thus, the NIN has better abstraction ability than traditional CNN. Moreover, the 1 × 1 ConvNets can adjust the tensor channels, making the structure design more flexible.

This research merges the inspirations from U-Net and NIN to construct the modified sky and ground segmentation network. It integrates the micro-networks in a U-shaped encoder-decoder structure. The proposed architecture is named “network in U-shaped network” (NI-U-Net). Figure 4 shows the overall structure of the proposed NI-U-Net, and Figure 5 shows the details. The black, orange, green, and blue squares refer to the input image (tensor), the 1 × 1 convolution-based micro-network (see Figure 5a), the stride-convolution-based scale reduction (see Figure 5b), and upsampling-convolution-based scale expansion (see Figure 5c), respectively. The green and blue braces indicate the encoder part and the decoder part, respectively. The scale reduction (green squares) and the micro-networks (orange squares) appear alternately in the encoder. The scale expansion (blue squares) and the micro-networks (green squares) in the decoder also appear alternately. The purpose of this design is that the scale changes are performed by stride-convolution and upsampling, and the micro-networks (with a deeper structure) conduct the feature abstraction.

The proposed sky and ground segmentation network has the following highlights. (1) The micro-network (Figure 5a) has N channels of input tensor, and the number of channels decreases to N/4 after the first 3 × 3 convolution. Then, the micro-network follows with two 1 × 1 convolutions. Finally, the micro-network applies another 3 × 3 convolution to restore the number of channels to N. Only the number of channels changes, while the image scale remains invariant. However, the first (far left) and last (far right) micro-networks are slightly different. The first micro-network inputs the image, and the number of channels is three. The first 3 × 3 convolution does not reduce it to N/4 but increases it to 32 channels. It is the first convolution over the entire network. The purpose is to increase the number of channels (this is a typical operation [57,58,59]). The output of the last micro-network is the prediction (output), so its activation is not “LeakyReLU” but “sigmoid” (for binary classification). (2) The scale reduction (the green square in Figure 4) uses convolution with a kernel size of 4 and a stride of 2 (Figure 5b). The convolution kernel is an integer multiple of stride, which can reduce the risk of artifacts [65]. (3) The scale increase uses the upsampling with a kernel of 2 (Figure 5c), which adopts nearest-neighbor interpolation. This research utilizes upsampling and convolution (rather than deconvolution) to avoid the risk of artifacts [65], which can increase the challenge in the training network.

The hyper-parameters of the pre-training are as follows. The pre-training network uses the Adam optimizer; the learning rate is set to 0.00001; the callback limitation of epochs is 50 epochs; the batch size is 32 samples per batch. The loss function applies the binary cross-entropy. The loss function translates the segmentation task to a binary classification task (sky or ground) for each pixel, which improves the segmentation to the pixel level.

This research adopts seven metrics to compare to the related research [5,26,31,34,35], including accuracy, precision, recall, Dice (F1) score, intersection over union (IoU) [34], misclassification rate (MCR) [31], and root mean squared error (RMSE). Equations (1)–(3) depict the mathematical definitions of IoU, MCR, and RMSE, respectively. Notably, some metrics are only for comparison to related studies rather than related to the training process. This research uses IoU and accuracy to indicate the performance, uses binary cross-entropy and RMSE to witness the loss trend during the training process. Furthermore, the related advanced studies adopt various metrics to discuss their performance. Thus, this research uses the same metrics to directly compare the performance (including precision, recall, Dice score, MCR, and RMSE).
(1)$$IoU=\frac{N_{TP}}{N_{TP}+N_{FP}+N_{FN}}$$
(2)$$MCR=\frac{N_{FP}+N_{FN}}{N_{p}}$$
(3)$$RMSE={\left(\frac{1}{m}\overset{m}{\underset{i=1}{\sum }}{\left(y_{i}-{\hat{y}}_{i}\right)}^{2}\right)}^{1/2}$$
where $N_{TP}$, $N_{FP}$, $N_{FN}$, $N_{p}$, m, $y_{i}$, and ${\hat{y}}_{i}$ refer to true-positive pixel number, false-positive pixel number, false-negative pixel number, total pixel number, category number, ground-truth label, and prediction, respectively.

Here is a brief description of the meaning and reasons for applied evaluation metrics. (1) The pixel-level semantic segmentation is a classification task on pixels. “Accuracy” is a very intuitive evaluation metric. (2) “Precision” refers to the proportion of the annotated sky pixels of the predicted sky pixels. “Precision” can characterize whether a large number of ground pixels are predicted as sky pixels. (3) “Recall” refers to whether the annotated sky pixels are also predicted as sky pixels. (4) Although “Accuracy” provides a very intuitive sense, which works properly with a balanced category distribution. The ratio of the sky and ground pixels is not strictly 1:1. Therefore, the “Dice score” is a more effective metric than the “Accuracy”. (5) “IoU” is a very important metric in image segmentation, and it is widely used in general image segmentation studies. (6) MCR is a per-pixel performance metric. Mihail et al. [31] used it to propose the benchmark for the Skyfinder dataset. (7) The above metrics are all pixel classification indicators, while the “RMSE” provides a metric based on the Euclidean distance. The smaller RMSE refers to a better result.

### 2.3. Conservative Annotation Strategy

A reasonable and efficient labeling strategy is essential for transferring the pre-trained network (Section 2.2) to the navigation visions of the planetary rover. There is no existing ground-truth sky and ground pixel-annotation for the navigation visions of the planetary rover, which is also a common challenge in the transfer learning tasks. Image labeling is a very complicated task. Besides, manual labeling reduces the reliability of the final result due to human errors. Figure 6 displays some difficult annotation regions in the Katwijk dataset, which have been highlighted with red frames. Therefore, this research proposes a novel labeling method, named the conservative labeling strategy, for the navigation visions of the planetary rover.

Figure 7 shows a result sample of the proposed conservative labeling strategy. Figure 7a displays a typical sample image from the Katwijk dataset. There are roughly four boundaries required to annotation. The boundaries “1”, “2”, and “3” are easy to locate (same location as the image borders) and provide high-constraints (higher pixel-ratio). The only difficult annotation happens to the skyline “4”. Figure 6 indicates that pixel-level annotation for skyline “4” is difficult, which can introduce significant human errors. In fact, the pre-training network achieves very accurate predictions in the Skyfinder dataset (discussed in Section 3.1). Therefore, the target of the proposed conservative labeling strategy for the Katwijk dataset is to perform special fine-tuning of the pre-trained network to fit the new scene, the navigation visions of the planetary rover.

The conservative labeling strategy preferentially guarantees the annotated skyline “4” located inside the corresponding image region (red or green region in Figure 7c). Then, the conservative labelling method pushes the annotated skyline to the natural skyline as close as possible without infecting too much annotation speed. (The annotated and natural skyline refer to the handcrafted label and the actual skyline, respectively.) Because of this conservative skyline selection criterion, the proposed strategy is named the conservative labeling strategy. In practical operation, the conservative labeling strategy takes about one minute per image. The labeling tool used in this research is Labelme [66]. Approximately 3% of the Katwijk dataset (150 images) has been annotated in this research.

Equations (4)–(13) brief the process of the weak supervision adopted in this research. It is notably that the technical terms of Domain, Task, Feature Space, and Marginal Probability Distribution can also be found in [67]. Domain is consisted of samples, while a single sample is represented using a single *Feature*
(X). All Features in a *Domain*
(D) consist of the *Feature Space*
(X). It is noteworthy that *Source Domain* ($D_{s}$) and *Target Domain* ($D_{t}$) of the transfer learning refer to the *Domain* of the pre-training and transfer-training *Sample Space*, respectively. Each Domain has two coefficients, the *Feature space* (X) and *Marginal Probability Distribution* (P(X)). The *Marginal Probability Distribution* (P(X)) refers to the marginal probabilities of the *Feature Space* (X). f(x) and $\Psi^{\prime }\left(x\right)$ refer to Task. For example, it can be the image segmentation. Transfer learning aims to achieve the f(x) in $D_{t}$ using the Prior Knowledge ($\Psi^{\prime }$) from $D_{s}$ [67].
(4)$$\Psi^{\prime }\left(x\right)=f\left(x\right)+\Phi \left(\left|x-\Lambda \right|\right)$$

Frosst and Hinton [68] claimed that a converged neural network should correspond to a Marginal Probability Distribution. In Equation (4), $\Psi^{\prime }$ and f refer to the same Task with different convergences. Convergence is a concept corresponding to the network. It is noteworthy that the Knowledge is decided by the Domain, and a converged network contains the Knowledge of the specific Domain. “$\Psi^{\prime }\left(x\right)$” applies the *Prior Knowledge* ($\Psi^{\prime }$) of $D_{s}$ to the x, and “f(x)” refers to the Knowledge of $D_{t}$. To distinguish these two *Knowledges*, this project uses the Prior Knowledge to correlate the Knowledge of the *Source Domain* ($D_{s}$), and the Converged Knowledge to correlate the Knowledge of the *Target Domain* ($D_{t}$).

Λ represents the sample space of $D_{s}$ that is only related to Marginal Probability Distribution of $D_{s}$, and “x−Λ” refers to the difference between any x and Λ in a board sense. “x−Λ” should not include any negative representation, so Equation (4) uses “|x−Λ|” to refer to absolute difference any x and Λ in a board sense. Therefore, Equation (4) relates the prediction from the Prior Knowledge and the Converged Knowledge of $D_{t}$ using Φ(|x−Λ|), where Φ refers that the difference between $\Psi^{\prime }$ and f is a function correlated to “|x−Λ|”. In another word, if $D_{s}$ and $D_{t}$ refer to the pre-training and transfer-training *Domains*, then Φ(|x−Λ|) should refer to the difference by straightforwardly applying the pre-trained model in the transfer-trained scenario.

During the beginning of the transfer learning process, the *Prior Knowledge* ($\Psi^{\prime }$) of $D_{s}$ should be different than the converged prediction (f) in $D_{t}$. Thus, Φ(|x−Λ|) does not equal to zero at the beginning. However, the essential of the transfer learning process should fine-tune the Prior Knowledge from the Marginal Probability Distribution of $D_{s}$ to $D_{t}$. Therefore, $\Psi^{\prime }$ and f should predict the same prediction, and Φ(|x−Λ|) should equal to zero.
(5)$$\dot{\Sigma }f\left(x\right)=f\left(X\right)$$

The above discussions are expressed from the view of a single sample, while the Prior Knowledge and Converged Knowledge should also be valid for the entire Domain. This research defines an Extension operator ($\dot{\Sigma }$) in a board sense, which refers to the process from a single sample x to X. Equation (5) depicts an example of the Extension from x to X. Equation (6) is accomplished by conducting $\dot{\Sigma }$ on Equation (4).
(6)$$\Psi^{\prime }\left(X\right)=f\left(X\right)+\Phi \left(\left|X-\Lambda \right|\right)$$

If the transfer learning is considered as an ongoing process, X refers to the intermedia feature space between $D_{s}$ and $D_{t}$. When X approaches to a closed point as Λ, Φ(|X−Λ|) should be a small value closed to zero so that the difference between $D_{s}$ and $D_{t}$ becomes small. It can be expressed as “$\Psi^{\prime }\left(X\right)≅f\left(X\right)$”. Therefore, if the annotation and learning is perfect, the *Prior Knowledge* represented by $\Psi^{\prime }$ should converge as same as f in $D_{t}$.

The above discussions assume the difference between the Prior Knowledge and the Converged Knowledge only comes from the difference between $D_{s}$ and $D_{t}$. Therefore, f can represent the ground-truth Marginal Probability Distribution which is the condition of supervised learning. However, the conservative annotated dataset is not fully supervised. Thus, the ground-truth f should be divided into two parts, $f^{\prime }$ and g. $f^{\prime }$ refers to the weak supervision from the conservative annotations, and g refers to the difference between f and $f^{\prime }$.
(7)$$\Psi^{\prime }\left(x\right)=\left[f^{\prime }\left(x\right)+g\left(x\right)\right]+\Phi \left(\left|x-\Lambda \right|\right)$$

Equation (7) replaces the f in Equation (4) with $f^{\prime }$ and g. g is straightforwardly correspond the unsupervised pixels in the conservative annotation. Notably, Ψ corresponds to the conservative annotations, and $\Psi^{\prime }$ corresponds to the completed annotations.
(8)$$\Psi \left(x\right)=\Psi^{\prime }\left(x\right)-g\left(x\right)=f^{\prime }\left(x\right)+\Phi \left(\left|x-\Lambda \right|\right)$$

Equation (8) moves the g(x) to the left side and pack with $\Psi^{\prime }$ as a new value, Ψ.
(9)$$\dot{\Sigma }\Psi \left(x\right)=\dot{\Sigma }\left(\Psi^{\prime }\left(x\right)-g\left(x\right)\right)=\dot{\Sigma }\left[f^{\prime }\left(x\right)+\Phi \left(\left|x-\Lambda \right|\right)\right]$$

Equation (9) performs $\dot{\Sigma }$ on Equations (4) and (8) in $D_{t}$, then achieves Equation (10).
(10)$$\Psi \left(X\right)=\Psi^{\prime }\left(X\right)-g\left(X\right)=f^{\prime }\left(X\right)+\Phi \left(\left|X-\Lambda \right|\right)$$

Now, Equation (11) assumes g(X) is a very small value to zero because of the prior knowledge and $\dot{\Sigma }$ from x to **X**.
(11)$$\Psi \left(X\right)=\Psi^{\prime }\left(X\right)-g\left(X\right)\;\approx \;\Psi^{\prime }\left(X\right)$$

Therefore, Equation (12) can be achieved from Equations (6), (10), and (11).
(12)$$f\left(X\right)+\Phi \left(\left|X-\Lambda \right|\right)=f^{\prime }\left(X\right)+\Phi \left(\left|X-\Lambda \right|\right)$$

Equation (13) eliminates Φ(|X−Λ|) from both sides in Equation (11). Equation (12) justifies the weak supervision from the theory aspect.
(13)$$f\left(X\right)\approx \;f^{\prime }\left(X\right)$$

To verify the above process and assumptions, this research uses the detailed experiments in Section 3.3.

### 2.4. Transfer Training Process: Sky and Ground Segmentation Network for the Navigation Visions of Planetary Rover

The transfer-training process is carried out on the *Prior Knowledge* of the pre-training process. This project proposes Hypothesis 1:
**Hypothesis 1.** Φ(|X−Λ|) *consists of two parts, the task-based loss (*$L_{task}$*) and the environment-based loss (*$L_{envi}$*). (See Equation (14))*
(14)$$\Phi \left(\left|X-\Lambda \right|\right)=L_{task}+L_{envi}$$

The Difference between the *Source Domain* ($D_{s}$) and the *Target Domain* ($D_{t}$) is the fundamental reason of Φ(|X−Λ|), which can be divided into two parts, the *Difference* caused by the *Task* change and the Difference caused by the Environment change. This project uses $L_{task}$ to characterize the loss related to *Task*, while $L_{envi}$ characterizes the loss related to environmental changes. Equation (6) can be transformed into Equation (15) according to Hypothesis 1, where $L_{task}$ is a function of *Task* and $L_{envi}$ is a function of *Environment*.
(15)$$\Psi^{\prime }\left(X\right)=f\left(X\right)+\Phi \left(\left|X-\Lambda \right|\right)=f\left(X\right)+\left(L_{task}+L_{envi}\right)$$

Equation (16) represents the pre-training process based on supervised learning.
(16)$$\underset{\Phi \left(\left|X-\Lambda \right|\right)\to 0}{\lim}\left[f\left(X\right)+\Phi \left(\left|X-\Lambda \right|\right)\right]=\underset{\left(L_{task}+L_{envi}\right)\to 0}{\lim}\left[f\left(X\right)+\left(L_{task}+L_{envi}\right)\right]$$

The transfer-learning process is a fine-tuning process based on the *Prior Knowledge* of the pre-training process. The Hypothesis 2 is:
**Hypothesis 2.** *If the pre-training has obtained superior sky and ground segmentation Prior Knowledge, it is considered that*$L_{task}$*has approached ZERO (see Equation**(17)).*
(17)$$L_{task}\;\approx 0$$

This research uses Hypothesis 2 to assume that the pre-trained model is already in a superior *Prior Knowledge* of recognizing the sky pixels, ground pixels, and skylines. The Φ(|X−Λ|) of using the pre-trained model in the planetary rover scene comes from the $L_{envi}$.

Therefore, Equation (17) can be substituted into Equation (15) to get Equation (18).
(18)$$\Psi^{\prime }\left(X\right)=f\left(X\right)+L_{envi}$$

Equation (18) depicts the same meaning as Equation (10), where g(X) refers to $L_{task}$, and $L_{envi}$ refers to Φ(|X−Λ|) in the transfer-training process.

It is essential to transfer the pre-trained achievement to the planetary rover scenario. Although the proposed NI-U-Net shows superior performance on the Skyfinder benchmark, Figure 3 and Figure 7a illustrate the distinctions between the Katwijk dataset and Skyfinder dataset, which also indicates the variant data distribution. However, the conservative labeling method can only generate limited samples. Thus, this research firstly performs data augmentation, then conducts the transfer learning.

This research adopts 22 augmentation schemes, including flip, brightness adjustment, contrast adjustment, crop, rotation, and color-channel shifting (see Figure A4 for more details). The augmentation increases the sample space from 150 to 3300 images. Notably, all augmented conservative data used for transfer training, and there are no validation and testing sets in the transfer training process. Validation and testing sets aim to evaluate the overfitting rate. However, transfer learning generally adopts to solve the problem with insufficient data. If there is enough data, there is no need for a transfer training strategy. Therefore, overfitting in transfer learning is inevitable to some extent. On the other hand, the essence of transfer learning is to fine-tune the pre-trained network with a small amount of data. Compared to dividing the validation and testing sets from a small dataset, using all available data for training can achieve better transfer training performance. The quantitative evaluation of the transfer learning result should use another part of accurately labeled data. Transfer learning directly loads the weight of the computation graph from the pre-trained network as the initializations. The learning rate sets to 0.00001. The starting point is closed to the eventual converging callback. Thus, the learning rate should be a closed value as in pre-training. The optimizer adopts Adam. The epoch callback sets to 300 epochs, and the batch size uses 15 images per batch.

This research further proposes the weakly supervised loss and accuracy for the conservative labeling method, which is named as the conservative binary cross-entropy ($loss_{bice}$) and the conservative binary accuracy ($acc_{bi}$). Equations (19) and (20) are the mathematical expressions of traditional binary cross-entropy and binary accuracy, respectively. The p, $y_{pred}$, label, N, and $N_{FN}$ refer to possibility, predictions, ground-truth labels, class number, and pixel number of false-negative, respectively. Notably, Equations (19) and (20) express the $loss_{bice}$ and $acc_{bi}$ with a function format (in programming). For example, the “$loss_{bice}\left(p\left(y_{pred}\right),\;label\right)$” (see Equation (19)) has the function declaration of “$loss_{bice}\left(argument1,\;argument2\right)$”, the first formal parameter is the “$p\left(y_{pred}\right)$”, and the second formal parameter is the “label”. Equation (20) follows the same pattern as Equation (19).
(19)$$loss_{bice}\left(p\left(y_{pred}\right),\;label\right)=-\frac{1}{N}\overset{N}{\underset{i=1}{\sum }}\left[\;label\cdot \log\left(p\left(y_{pred}\right)\right)+\left(1-label\right)\cdot \log\left(1-p\left(y_{pred}\right)\right)\right]$$
(20)$$acc_{bi}\left(p\left(y_{pred}\right),\;label\right)=\frac{N_{TP}+N_{TN}}{N_{TP}+N_{TN}+N_{FP}+N_{FN}}$$

Pre-training is a binary classification task, but transfer learning becomes an “incomplete” multi-classification task. There are three types of pixels in conservative annotations (sky (red), ground (green), and unannotated (black) pixels in Figure 7). The sky and ground pixels are annotated “easy” pixels, while the unlabeled pixels are unannotated “hard” pixels. The transfer learning in this research can only rely on the “easy” pixels rather than all pixels. Equations (21) and (22) are the proposed conservative (binary) cross-entropy ($loss_{con}$) and conservative (binary) accuracy ($acc_{con}$).
(21)$$loss_{con}=loss_{bice}\left[\left(y_{pred}*mask_{s}\right),label_{s}\right]+\alpha *loss_{bice}\left[\left(\left(1-y_{pred}\right)*mask_{g}\right),label_{g}\right]$$
(22)$$acc_{con}=\frac{1}{2}*\left[\left(\frac{acc_{bi}\left(y_{pred},label_{s}\right)-\frac{N_{s}}{N_{p}}}{1-\frac{N_{s}}{N_{p}}}\right)+\beta *\left(\frac{acc_{bi}\left(\left(1-y_{pred}\right),label_{g}\right)-\frac{N_{g}}{N_{p}}}{1-\frac{N_{g}}{N_{p}}}\right)\right]$$
where $mask_{s}$, $label_{s}$, $mask_{g}$, $label_{g}$, α, β, $N_{s}$, and $N_{g}$ refer to conservative sky-mask, conservative sky-labels, conservative ground-mask, conservative ground-labels, weight for $loss_{con}$, weight for $acc_{con}$, sky-pixel number, and ground-pixel number, respectively.

Algorithm A1 in Appendix A as well as Equations (21) and (22) explain the procedure of calculating the $loss_{con}$ and $acc_{con}$ in one backpropagation (the following eight steps).

(1)The sky and ground segmentation network inputs a batch of images and outputs a prediction ($y_{pred}$), while the corresponding conservative label is $label_{con}$. The $label_{con}$ can be divided into the sky, ground, and unannotated pixels using two thresholds ($\lambda_{s}$ and $\lambda_{g}$). (Notably, all s and g subscripts indicate sky and ground.)(2)This research calculates the number of sky ($N_{s}$) and ground ($N_{g}$) pixels in $label_{con}$, while the $N_{s}/N_{p}$ and $N_{g}/N_{p}$ refer to sky and ground pixel ratio in $label_{con}$, respectively.(3)This research produces a $mask_{s}$ and $mask_{g}$, where $mask_{s}$ has all conservative annotated sky pixels with value one and others with value zero, where $mask_{g}$ has all conservative annotated ground pixels with value one and others with value zero.(4)This research conducts the pointwise multiplications between $y_{pred}$ and $mask_{s}$ to achieve the filtered conservative sky pixel prediction (see “$\left(y_{pred}*mask_{s}\right)$” in Equation (21)), which only remains the predictions at the exact locations of the conservatively annotated sky pixels.(5)This research uses value one to pointwise-subtract the $y_{pred}$ (the “$\left(1-y_{pred}\right)$” in Equation (21)) because the situation of sky and ground should be opposite. This research conducts a similar process (as step (4)) to achieve a filtered conservative ground pixel prediction (see “$\left(\left(1-y_{pred}\right)*mask_{g}\right)$” in Equation (21)).(6)This research generates the $label_{s}$ and $label_{g}$ to indicate the annotated sky and ground pixels only. The $label_{s}$ has sky pixels with value one and others with zero. The $label_{g}$ has ground pixels with value one and others with zero.(7)This research calls the $loss_{bice}$ function (Equation (19)) with the input of step (4) and step (6) to achieve the conservative sky cross-entropy. This research calls the $loss_{bice}$ function (Equation (19)) with the input of step (5) and step (6) to achieve the conservative ground cross-entropy. This research further adds a weight parameter “α” to balance the two cross-entropies.(8)Equation (22) calls the $acc_{bi}$
function (Equation (20)) with the input of step (4) and step (6) to achieve the conservative sky accuracy. Equation (22) calls the $acc_{bi}$ function (Equation (20)) with the input of step (5) and step (6) to achieve the conservative ground accuracy.

## 3. Results and Discussion

The operating system of this research is Ubuntu 18.04, the deep learning environment is TensorFlow 2.1, GPU support is CUDA 10.1, and the programming language is Python 3.6. The hardware configuration is i7-7700 CPU, RTX1080 graphics card, 32 GB memory.

### 3.1. Pre-Trained Results in the Skyfinder Dataset

Figure 8 shows the binary cross-entropy loss curves (Figure 8a), accuracy curves (Figure 8b), IoU curves (Figure 8c), and Dice score curves (Figure 8d) of the proposed NI-U-Net in the pre-training process. (i) The curves are relatively smoothed, indicating the pre-training process is very stable. These curves are relatively smooth, indicating the division ratio (of training, validation, and testing sets) and the batch size setting are effective. A proper batch size can avoid the generation gap and provide an efficient gradient for backpropagation [69]. (ii) There is no sign of overfitting. The training and validation losses keep decreasing; meanwhile, the training and validation accuracy both keep increasing. (iii) There is no sign of underfitting either. The training and validation accuracies achieve a high value of over 98%. Thus, the Skyfinder dataset used in this research has sufficient diversity. The proposed sky and ground segmentation network has good data fitting capability.

The curves in Figure 8 appear similar learning trends, which indicate a possible risk of data contamination. This research eliminates this concern using Figure A5, which is the corresponding curves from a re-implemented U-Net. Notably, the training/validation/testing sets for Figure 8 and Figure A5 are the same. The curves in Figure A5 do not appear similar learning trends, which indicate the data contamination does not occur. Furthermore, Figure 8 appears the crosses between the training and validation curves, which is another evidence. Besides, the re-implementation for the U-Net has referenced the codes in [70,71], which both have high stars in GitHub.

Table 1 provides the loss, accuracy, IoU, and Dice score of the training, validation, and testing sets of the pre-training process. This research also provides the results from U-Net experiments in Table 1. The proposed NI-U-Net is a highly modified network based on the general U-Net. It is straightforward to compare the results of the proposed NI-U-Net to the original U-Net.

Table 2 shows the results of the pre-training process on the Skyfinder dataset. This research quantitatively compares to the state-of-the-art by adopting seven metrics. The pre-trained sky and ground network achieved the best results on the open benchmark, the Skyfinder dataset [34]. The “NI-U-Net” row (in Table 2) refers to the results from the proposed network in the pre-training process. Notably, Hoiem, Tighe, Lu, Mihail, Song, Place, and Nice are not tested at the same images as the “U-Net” and “NI-U-Net” (because their code is not provided). This research lists their results as additional evidence. However, the “U-Net” refers to the experimental results from the same training, validation, and testing sets as the “NI-U-Net”. Table 2 depicts the proposed NI-U-Net achieves the best results among all seven metrics. Notably, the U-Net appears a higher value in RMSE compared to “Nice (2020)” and NI-U-Net. The prediction values in the binary classification task cannot directly influence the classification. Regarding the accuracy, any prediction value over “0.5” is decided as the one-hot classification of value “1”.

Figure 9 shows the qualitative performance of the pre-trained network in some typical scenes of the Skyfinder dataset. This research displays the same scenes as in the related research [5,26,31,34,35] to qualitatively highlight the improvements in this research.

The last column in Table 2 depicts the retained amount of the images in the Skyfinder dataset. This research retains the most Skyfinder dataset compared to all the benchmarks, which means the pre-training data is more challenged than the benchmarks (10% to 50% more). Many of them are very difficult in terms of visual conditions because of illumination, weather, or noises (see Section 2.1.1 and Figure 3).

However, the pre-trained network showed excellent performance in the difficult scenes. Figure 10 depicts some example results from difficult scenes. (i) A widespread challenge in vision-based applications is the backlight. The pre-trained network recovered the invisibility of the backlight at a relatively high level. (ii) The “cloud and contrast” column indicates a high contrast situation from a part of the sky. The pre-trained network can identify the correct ground pixels. (iii) The “dark and noise”, “illumination”, “night” columns indicate the scenes with low illumination and high noises. The sky and ground pixels were detected precisely, which show the robustness for noises and illumination. (iv) The “fog” column shows a blur around sky and ground skyline. It is very challenged even for human vision, while the pre-trained network accurately detected the sky and ground pixels.

### 3.2. Intermedia Results of Using Pre-Trained Network in the Katwijk Dataset

Figure 11 illustrates the intermedia results of straightforwardly using a pre-trained network in the Katwijk dataset without transfer training process. Figure 11a shows three random scenes from the Katwijk dataset, and Section 3.2 and Section 3.3 use them as an example to compare the performance between the pre-trained network and transfer-trained NI-U-Net. Figure 11b refers to the conservative annotations of the corresponding scene in Figure 11a. Figure 11c illustrates the results of using the pre-trained network in the Katwijk dataset. The pre-trained network has achieved the Prior Knowledge of sky and ground segmentation according to the following points. The intermedia results support the discussion in Section 2.1.1, which is the pre-training process shares a similar fundamental logic as in the navigation vision of the planetary rovers.

(1)The pre-trained network has generally found the sky and ground area (the black and white predictions approximately gathered at corresponding image region).(2)The pre-trained network also roughly identified the skyline between sky and ground (a shallow trace at the top of the image region).

However, the results of the pre-trained network do not accomplish the expected results for the navigation vision of the planetary rovers. There are still significant wrong predictions in both sky and ground areas, which verifies the assumption of $L_{envi}$ (in Section 2.4) from the Feature Space and Marginal Probability Distribution between the Skyfinder dataset and the Katwijk dataset.

### 3.3. Final Results in the Katwijk Beach Planetary Rover Dataset

The annotation methods in most annotation software can be divided into three categories, the bounding-box, object tags, and pixel-annotation [72]. Although the bounding-box notation can satisfy the mathematical logic in Section 2.3, it discards the unique distributions of the sky and ground pixels. The unique distributions of the sky or ground pixels are normally clustered or connected image regions. The tag notation is obviously not suitable for this research because all scenes have sky and ground tags. The pixel-annotation method consumes too much time and human resources [46]. The proposed conservative annotation method combines the ideas of bounding-box and pixel-annotation. From the perspective of efficiency, the conservative dataset can be annotated quickly and efficiently. The average annotation time is about one to two minutes per image in the Katwijk dataset. Figure 12 and Figure A4 display the results of the conservative annotation method and the following data augmentations. The conservatively annotated image is about 3% of the random selected frames from the navigation video stream of the planetary rovers.

The transfer training process achieved excellent results with very limited manual annotations, which indicates that the proposed framework based on the weak supervision and transfer learning has outstanding practicality and robustness for the navigation visions of planetary rovers. Figure 11d displays the predictions from the transfer-trained network with the corresponding scenes in Figure 11a. Table 3 introduces the experimental results of the transfer-trained network with the conservative loss ($loss_{con}$) and conservative accuracies ($acc_{con}$). This research is dedicated to providing assistance for the semantic landmarks detection of the planetary rovers. The sky and ground segmentation network can provide efficient semantic tags for all landmarks appearing in the navigation visions. Figure A6 utilizes red and green frames to highlight some example landmarks. The sky and ground segmentation can help to eliminate some wrong landmarks. Furthermore, the blue frame highlights the skyline between sky and ground, which can provide rich landmarks such as shadows, rocks, stones, traces, clouds, hills, mountains, etc.

Figure 13 depicts the training records of the proposed NI-U-Net in the transfer training process. The accuracy curve converges at about 30th epochs, which is earlier than the loss curve (until 200th epoch). The different converge speed supports the discussion about the continuous and discontinuous variable loss and accuracy in Section 3.1. Moreover, the overall transfer training process does not occur the over-fitting because the accuracy and loss curves stay flat. Furthermore, the accuracy keeps at a high level, so the underfitting does not occur. Besides, there are some pikes in both loss and accuracy curves. The training strategy in this research is the batch-based random gradient descent, and it is common to occur some unstable “vibrations.” However, the importance is that the curve keeps in a proper general trend. The loss curve keeps decreasing, and the accuracy curve keeps increasing. The conservative accuracy can only be used as an indicator to measure the training ability of conservatively annotated labels in the transfer training process. The “difficult” pixels (closed to the skylines) cannot be used for calculating the traditional accuracy.

This research utilizes the full-manually annotated images from the Katwijk dataset to evaluate the network of transfer learning. There are different 30 images (the Katwijk dataset) that have been annotated as evaluation scenes. Notably, the ratio of evaluation and transfer training images is 1:5. This research then conducts the same augmentations (the same as in Section 2.4) to the evaluation images to 660 images. Thus, these (fully annotated) 660 images become the evaluation set for the transfer learned NI-U-Net. Table 4 indicates that the evaluation cross-entropy, accuracy, IoU, and Dice score are 0.0916, 99.269%, 99.256%, and 99.626%, which stays at a high level. The evaluation loss (in Table 3) and evaluation loss (in Table 4) are two different values (1.0714 and 0.0916), which verifies that the specific value of the loss does not directly relate to the network performance (discussed in Section 3.1).

The all above discussions have verified the feasibility of using the proposed weak supervision and transfer learning framework to handle the sky and ground segmentation problem in the planetary rover’s navigation vison. This is also a verification of the assumption in Section 2.3.

This research attached a demo video that integrates the sky and ground segmentation network (from the transfer learning) into the navigation visions of the Katwijk planetary rover dataset to further illustrate the training results. This research applies the inference time and framerate as indicators for the real-time analysis. In the GPU-support of RTX1080, the proposed NI-U-Net achieves about 0.026 s per frame for the average inference time and 38 frames per second (fps) for the framerate. Figure A8a depicts all inference times of the 30 evaluation scenes. (The evaluation scenes in this paper refer to the 30 images used in Table 4). The framerate and inference time are far sufficient for the real-time requirements of navigation video. Furthermore, the video speed of the attached demo video is eight fps. The processing speed of the network proposed in this research is about five times higher than the fps in the demo video. Furthermore, this research uses OTSU image segmentation as a comparison. The timing curves are depicted in Figure A8b, and the ROC, POR, and AUC curves are illustrated in Figure A9. Although OTSU has faster processing speed than NI-U-Net, the performance is not actually acceptable (see Figure A10).

This research also utilizes re to visualize the possibility of the transfer-trained NI-U-Net in some example images from the NASA album [56]. Figure A7 displays the sky and land segmentations corresponding to the view in Figure A1. It is noteworthy that Figure A7 does not mean the transfer trained NI-U-Net can be directly used for [56], but it has some robustness for some scenes. Different planetary environments correlate to significantly variant environment distributions, it is impossible to expect one model can take care of all planetary missions. Besides, the aim of this research is to propose a visible framework for the sky and ground segmentation in the navigation vision system with the planetary rovers. According to the results in Section 3.3 and the visualization in Figure A7, the proposed framework has the ability to handle similar missions as a framework.

## 4. Conclusions and Future Works

This research proposes a sky and ground segmentation (sky and ground segmentation) framework based on weak supervision and transfer learning for the navigation visions of the planetary rover, which can improve the accuracy and efficiency of semantic landmarks detection in the practical scenario. This research designed a new convolution-only sky and ground segmentation architecture, which overcomes the blurry prediction, information loss, and vanishing gradient that occurs in existing solutions. The pre-training achieves the best results on the open benchmark (the Skyfinder dataset) among all seven metrics compared to state-of-the-art. This research further proposes a conservative labeling method, which successfully transfers the pre-trained state-of-the-art segmentation network to the planetary rover vision. This research appears to be the first successful attempt at sky and ground segmentation for the navigation visions of the planetary rover.

This research is an E2E high-performance solution in practical scenarios and can meet real-time requirements compared with related work. The accuracy, precision, recall, Dice score, MCR, RMSE, and IoU are 99.232%, 99.211%, 99.221%, 99.104%, 0.0077, 0.0427, and 98.224%, respectively. This research also shows promising results in some Mars rover images (see Figure A1 and Figure A7). However, the proposed NI-U-Net has not been tested in the onboard devices. To transfer the proposed framework to the onboard device requires a few more steps. (1) The support deep learning library (normal TensorFlow) needs to transfer to the lite version (like TensorFlow-Lite). (2) The loss function and evaluation metrics need further customizations. (3) To run in onboard devices, the proposed NI-U-Net requires more optimization for the onboard system, hardware, and software. This can be a great future work since this is not the focus of this research.

This research can be widely used in semantic segmentation, target detection, and rover autonomy tasks of planetary exploration vision. The proposed conservative labeling method (of the weak supervision) can help transform many advanced achievements with laboratory conditions into practical applications. Furthermore, even though the weak supervision process is a binary classification task, the proposed weak supervision method based on the conservative labeling method can be widely applied to multi-classification application scenarios.

## Figures and Tables

**Figure 1 sensors-21-06996-f001:**
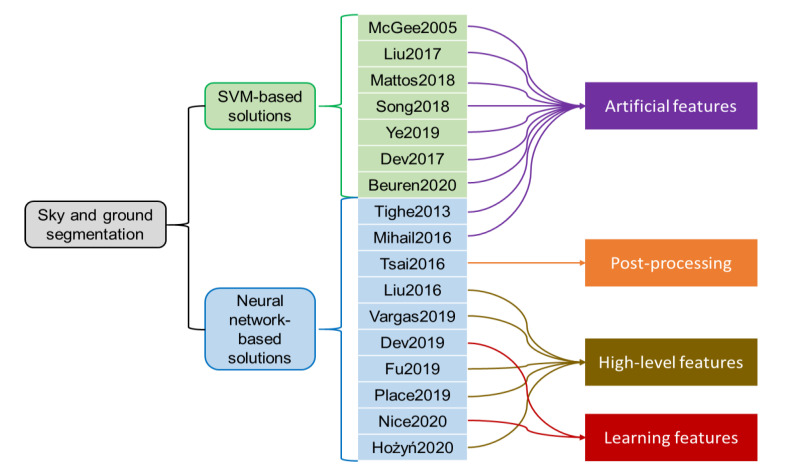
The group of the related works. McGee2005, Liu2017, Mattos2018, Song2018, Ye2019, Dev2017, Beuren2020, Tighe2013, Mihail2016, Tsai2016, Liu2016, Vargas2019, Dev2019, Fu2019, Place2019, Nice2020, and Hożyń2020 refer to [2,10,12,23,24,25,26,27,28,31,32,33,34,35,36,39], respectively.

**Figure 2 sensors-21-06996-f002:**
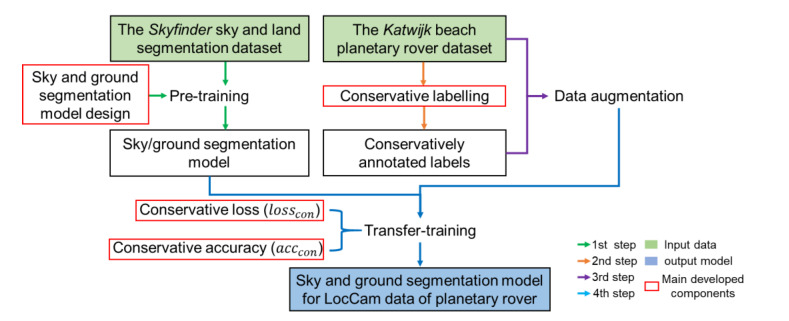
The overall process of the proposed framework. The red frames refer to the main innovations in this research. Green and blue frames refer to the inputs and outputs of the framework. LocCam refers to the localization camera of the planetary rovers.

**Figure 3 sensors-21-06996-f003:**
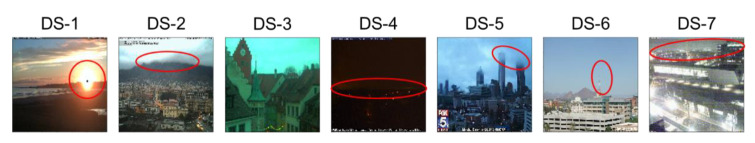
The difficult scenes in the Skyfinder dataset. The red circles highlight the difficult visible situation. “DS” refers to the difficult scene. DS-1, DS-2, DS-3, DS-4, DS-5, DS-6, and DS-7 refer to the strong backlight, heavy fog occlusion, color-channel shift, extremely low illumination, heavy cloud occlusion, truss structure, and heavy noises, respectively.

**Figure 4 sensors-21-06996-f004:**
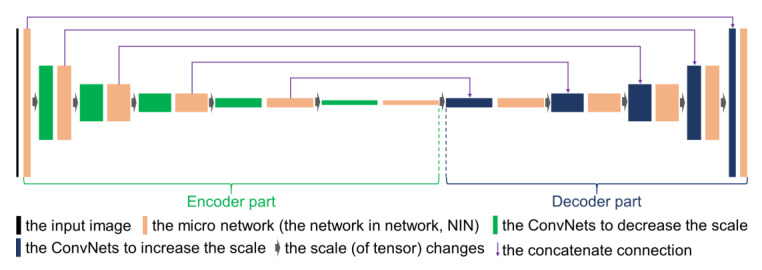
The proposed sky and ground segmentation network. The green and blue braces refer to the encoder and decoder parts, respectively. The dark vertical thick line on the left refers to the input RGB image. The green, orange, and blue squares refer to the stride-based ConvNet, NIN structure, and upsampling-based ConvNet, respectively. The details have been illustrated in Figure 5 for the green, orange, and blue squares. The grey arrows highlight the dimension change in the height and width axis of the tensors. The purple arrows refer to the concatenate layer to connect the encoder and decoder layers.

**Figure 5 sensors-21-06996-f005:**
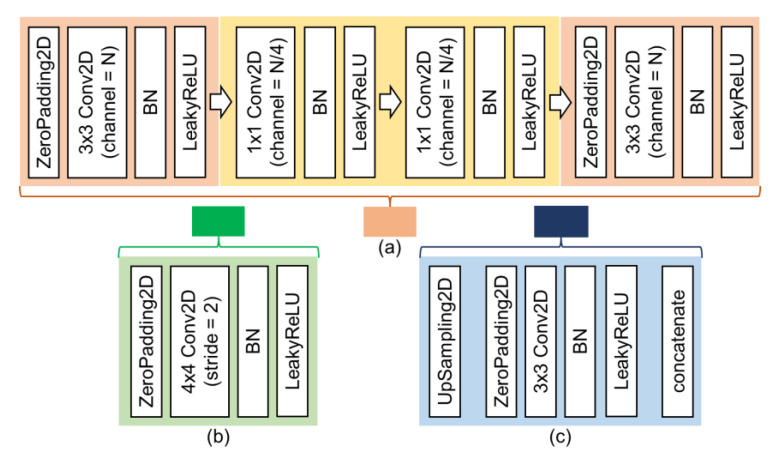
The detailed network design for the proposed sky and ground segmentation network. The green, orange, and blue squares refer to the network structure in Figure 4, respectively. “ZeroPadding2D” refers to the 2D padding layer using the value zero. “3 × 3 Conv2D (channel = N)” refers to the 2D ConvNet using a kernel size of 3, and the number of output channels equals value N. “BN” refers to the batch normalization layer. “LeakyReLU” refers to the activation layer using the LeakyReLU function. (**a**) refers to the NIN structure applied in this research. (**b**) refers to the stride-ConvNet in the encoder part. (**c**) refers to the ConvNet for dimension increase.

**Figure 6 sensors-21-06996-f006:**
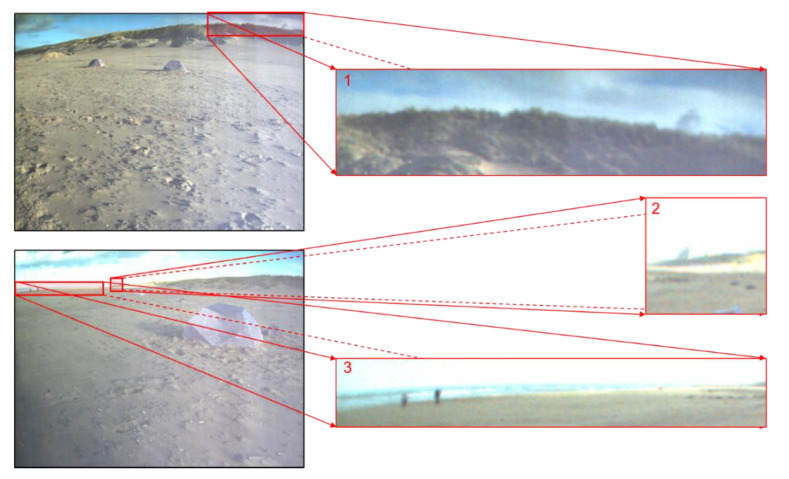
The difficult sample regions for manual annotation in the Katwijk beach planetary rover dataset. The red frames highlight the difficult sample regions. “1” refers to a broken edge, which has a very complicated structure. “2” refers to a rock target, which is difficult to identify the skyline part. “3” refers to a high reflection target, and its skyline part is also difficult to identify.

**Figure 7 sensors-21-06996-f007:**
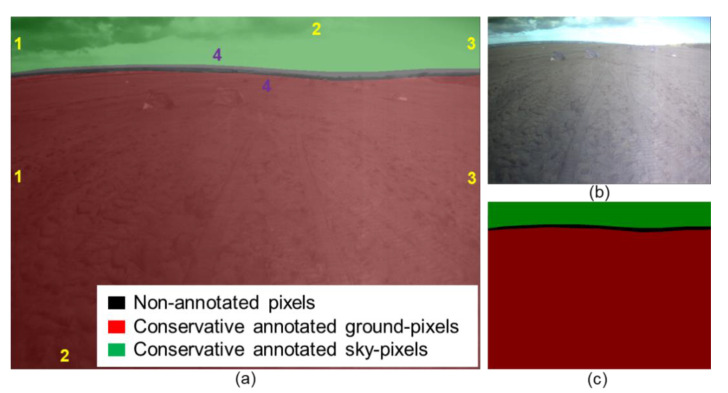
The proposed conservative labeling method. “1”, “2”, “3”, and “4” refer to the index of the boundaries. Gold “1”, “2”, and “3” refer to the easy-annotated boundaries, and purple “4” refers to the difficult-annotated boundaries. (**a**) adds the transparent conservative label to the original image. (**b**) refers to the original image. (**c**) refers to the conservative label.

**Figure 8 sensors-21-06996-f008:**
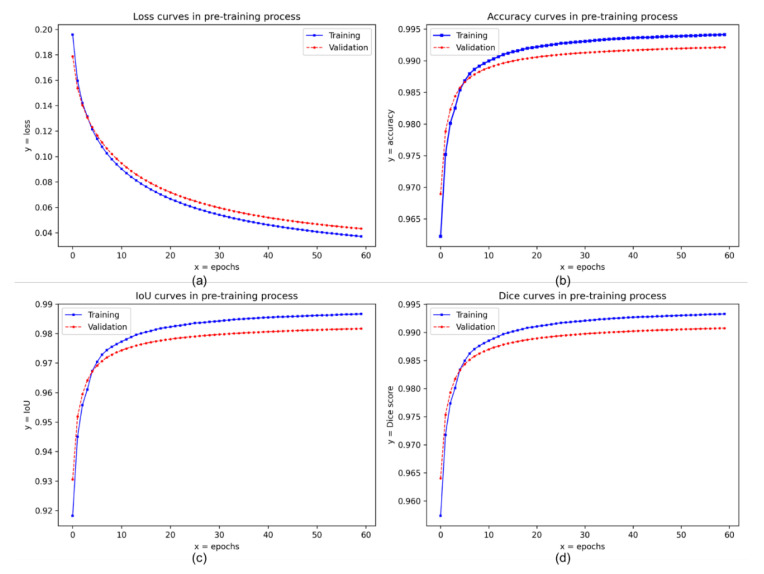
The loss (binary cross-entropy), accuracy, IoU, and Dice score records of the pre-training process in training and validation set (the Skyfinder dataset). (**a**–**d**) refer to the loss (binary cross-entropy), accuracy, IoU, and Dice score, respectively.

**Figure 9 sensors-21-06996-f009:**
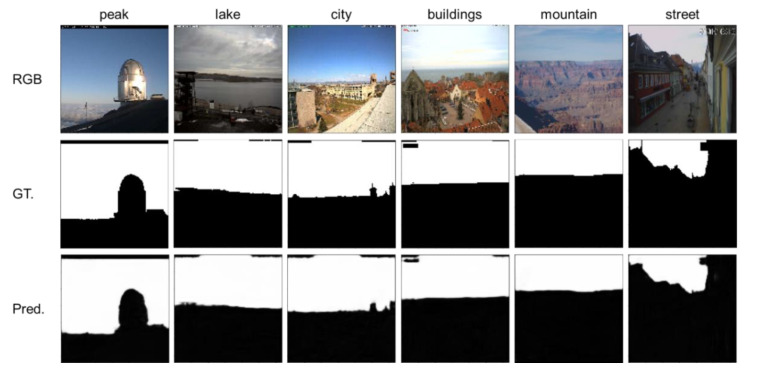
The result of the pre-trained network in normal visual conditions of the Skyfinder dataset. “RGB”, “GT.”, and “Pred.” refer to the input source images, the ground-truth annotations, and the predictions of the proposed NI-U-Net, respectively. “peak”, “lake”, “city”, “buildings”, “mountain”, and “street” refer to some example scenes in the Skyfinder dataset.

**Figure 10 sensors-21-06996-f010:**
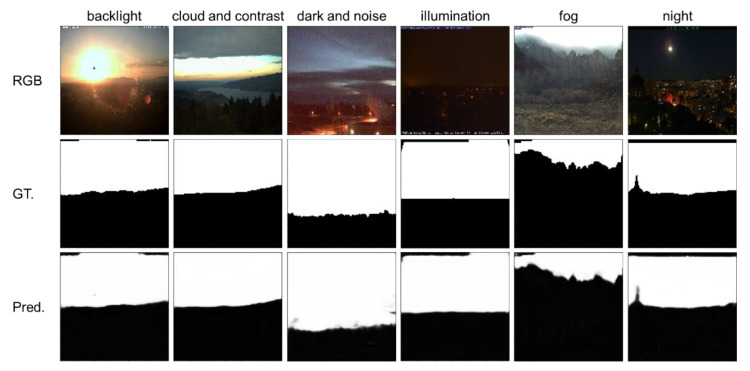
The sky and ground segmentation results of the pre-trained network in many difficult scenes. “RGB”, “GT.”, and “Pred.” refer to the input source images, the ground-truth annotations, and the predictions of the proposed NI-U-Net, respectively. “backlight”, “cloud and contrast”, “dark and noise”, “illumination”, “fog”, and “night” refer to some example difficult scenes in the Skyfinder dataset.

**Figure 11 sensors-21-06996-f011:**
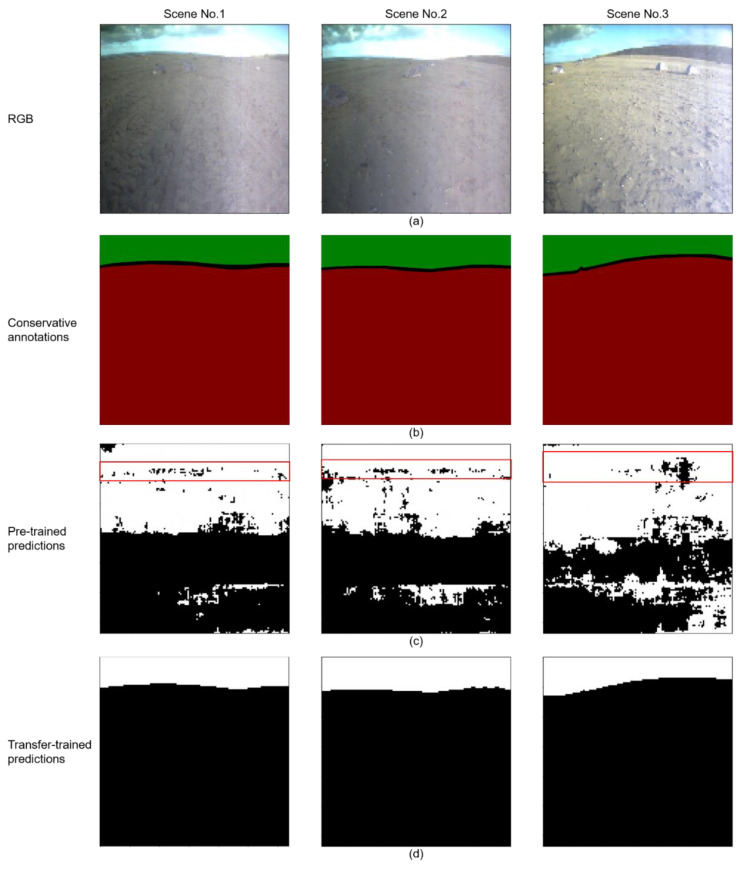
The visualizations of pre-trained and transfer-trained networks for three random scenes from the Katwijk dataset. (**a**) refers to the RGB input images from the Katwijk dataset. (**b**) refers to the corresponding conservative annotations from Section 2.3. (**c**) refers to the prediction from the pre-trained network in the Katwijk dataset. The red frames highlight the approximate skylines, and the white and black pixels have found the rough sky and ground regions. (**d**) refers to the results of the transfer-trained network. It is noteworthy that each column only correlates to a single scene.

**Figure 12 sensors-21-06996-f012:**
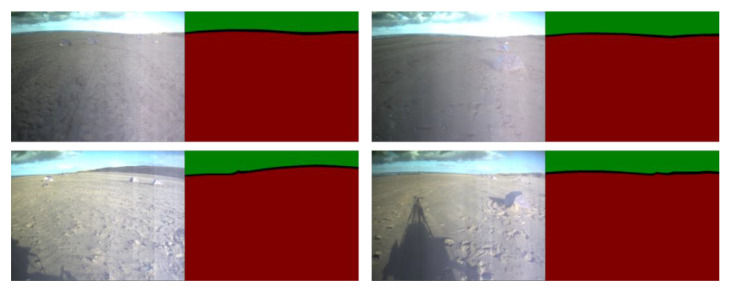
The conservative labels for the navigation visions of the planetary rover. The red, green, and black regions refer to the ground, sky, and un-annotated pixels, respectively. The left figures come from the Katwijk dataset [47].

**Figure 13 sensors-21-06996-f013:**
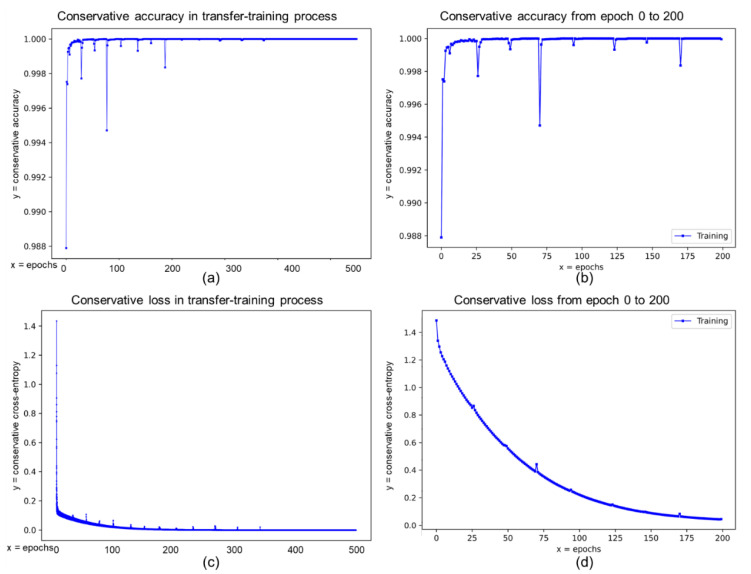
The conservative loss ($loss_{con}$) and conservative accuracy ($acc_{con}$ )) records of the transfer-training process in training and validation set (augmentation data). (**a**,**b**) refer to the epoch-wised $acc_{con}$ in the transfer-training process. (**c**,**d**) correspond to the epoch-wised $loss_{con}$ in the transfer-training process. (**b**,**d**) are the enlarge figures between epoch 0 and 200 from (**a**,**c**).

**Table 1 sensors-21-06996-t001:** The loss, accuracy, RMSE, and IoU results of the pre-trained sky and ground segmentation network.

Networks	Item	Loss	Accuracy	IoU	Dice Score
	Unit	NA	%	%	%
U-Net	Training set	0.0268	98.844	97.625	98.798
	Validation set	0.0455	98.281	96.113	98.018
	Testing set	0.0453	98.285	96.142	98.033
NI-U-Net	Training set	0.0372	99.411	98.664	99.327
	Validation set	0.0433	99.212	98.169	99.076
	Testing set	0.0427	99.232	98.224	99.104

The symbol “NA” and “%” refers to non-applicable and percentage, respectively. The “U-Net” refers to the related research [57]. The “NI-U-Net” refers to the pre-trained results from this research.

**Table 2 sensors-21-06996-t002:** The pre-trained sky and ground segmentation network results evaluated in the open benchmark (the Skyfinder dataset).

Metrics	Accuracy	Precision	Recall	Dice Score	MCR	RMSE	IoU	Retained Amount
**Unit**	%	%	%	%	%	**NA**	**NA**	**Images**
Hoiem (2005)	NA	NA	NA	NA	22.83	NA	NA	18,000
Tighe (2010)	NA	NA	NA	NA	19.51	NA	NA	18,000
Lu (2014)	NA	NA	NA	NA	25.08	NA	NA	18,000
Mihail (2016)	NA	NA	NA	NA	12.96	NA	NA	NA
Song (2018)	NA	NA	NA	NA	96.30	NA	NA	18,000
Place (2019)	NA	NA	NA	NA	58.90	NA	0.8300	60,000
Nice (2020)	NA	94.6	96.5	95.2	NA	0.063	NA	38,000
U-Net	98.285	98.464	98.094	98.033	0.0172	0.131	0.9614	70,000
**NI-U-Net**	**99.232**	**99.211**	**99.221**	**99.104**	**0.0077**	**0.0427**	**0.9822**	70,000

The bold contexts refer to the best value in the corresponding metric. “Hoiem”, “Tighe”, and “Lu” refer to the baseline results provided in [26], respectively. Mihail, Song, Place, and Nice refer to the recent advanced results in [26,31,34,35], respectively. The “U-Net” refers to the results from the re-implemented U-Net. The “NI-U-Net” refers to the pre-trained results from this research. The symbol “NA” refers to either not applicable or not available.

**Table 3 sensors-21-06996-t003:** The experimental results of the transfer learned NI-U-Net.

$\mathrm{Conservative}\;\mathrm{Loss}\;(loss_{con})$	$\mathrm{Conservative}\;\mathrm{Accuracy}\;(acc_{con})$	$\mathrm{Conservative}\;\mathrm{Accuracy}\;(acc_{con})\;\mathrm{of}\;\mathrm{Sky}\;\mathrm{Pixels}$	$\mathrm{Conservative}\;\mathrm{Accuracy}\;(acc_{con})\;\mathrm{of}\;\mathrm{Ground}\;\mathrm{Pixels}$
1.5606375 × 10^−7^	~100.0%	~100.0%	~100.0%

The $loss_{con}$ and $acc_{con}$ refer to Equations (21) and (22), respectively. “~100%” refers to the value very close to 100%.

**Table 4 sensors-21-06996-t004:** The results of the (fully annotated) evaluation images for the transfer learning.

Evaluation Loss	Evaluation Accuracy	Evaluation IoU	Evaluation Dice Score
0.0916	99.269%	99.256%	99.626%

## Data Availability

The data presented in this study are openly available in Multimodal Vision Research Laboratory at https://doi.org/10.1109/WACV.2016.7477637 and ESA Robotics Datasets at https://doi.org/10.1177/0278364917737153, reference number [31] and [47].

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
