# Peer review of "Sky and Ground Segmentation in the Navigation Visions of the Planetary Rovers"

_sensors, 2021, doi:10.3390/s21216996_

Round 1

Reviewer 1 Report

The authors exposed a novel method to segment images into sky and non-sky. The method uses ConvNet, Neural in Network (NIN) structure, in a U-net structure. Moreover, to lessen the problem of miss segmentation, they use a conservative annotative strategy that leaves ambiguous regions of an image (here at the boundary between sky and ground) non annotated.

The articles mentioned several algorithms and grouped their problem and limitation in a summary figure. It will also be interesting if the authors could have information about the computational complexity of the other methods , and their running time to compare the benefits of their new method to those mentioned.

Several metrics are reported to expose the quality of the author implementation. It is, however not clear why such metrics have been chosen. Furthermore, when looking at table 2, many metrics are not reported in previous work, making the comparison to other algorithms difficult.

In the sky finder dataset, specific images ("bad" images) are eliminated because of overexposure, saturation, heavy fog, strong backlight, etc. In a real world scenerio, such images may be encountered. I am wondering how such images could be detected, and if not, which segmentation would be given by the algorithm on such images. I would, for example, be curious about the segmentation on the images shown in A1.

In the conclusion and future work, I felt a bit short on the approach. The work is presented from a computer vision perspective. It would be interesting to discuss to bio-inspired approaches to sky-ground segmentation. For example, the following articles may be of interest:

- Freas CA, Whyte C, Cheng K. Skyline retention and retroactive interference in the navigating Australian desert ant, Melophorus bagoti. J Comp Physiol A. 2017;1–15.

- Towne WF, Ritrovato AE, Esposto A, Brown DF. Honeybees use the skyline in orientation. J Exp Biol. 2017;220(13):2476–85. Doi:10.1242/jeb.160002

- Stone T, Differt D, Milford M, Webb B. Skyline-based localisation for aggressively manoeuvring robots using UV sensors and spherical harmonics. In: 2016 IEEE International Conference on Robotics and Automation (ICRA). IEEE; 2016. p. 5615–22.

- Müller MM, Bertrand OJN, Differt D, Egelhaaf M. The problem of home choice in skyline-based homing. Graham P, editor. PLoS One. 2018;13(3):e0194070. Doi:10.1371/journal.pone.0194070

Minor comments:

- Lines 136 – 146: The reasoning for this pipeline is not really clear. It would help to add sentences about the motivation behind the pipeline.

- Lines 166: “70,000 images from 53 scenes” Why all 53 scenes were chosen and not a portion of it in order to test for generality in there. 53 locations in the data set seem arbitrary.

- Lines 167: “the pre-training in this research is more challenged” It can also then better generalise to the rest of the dataset, but may fail to generalise to other dataset.

- Line 207-209: Why does a video stream format not fit the aim? Why one could not convert them first to another format in order reach the aim?

- Line 314: “Manual labeling reduces the reliability of the final result due to human errors”. This point can be reduced by using concensus between multiple labellers.

- Line 348: Why only 150 images were labelled. Since it takes only one minute per image, it seems that this number could be increased easily.

- Line 362-363: ". Convergence is a concept correspond to networks, while this

project uses Knowledge to correspond the concept of Domain which refers to the same thing

as convergence. “ The sentence is difficult to understand because it contains multiple concept.

- Line 530-536: It is not clear what one should conclude from it. Would it be better to report on the re-implementation of U-NET based on unit testing and benchmark testing instead of the number of start in GitHub?

Reviewer 2 Report

This is a well-presented paper for sky-ground segmentation with transfer learning and weak supervision. The conservative annotation is a good idea for transfer learning. However, the data augmentation is not clearly described, please expand it clearly. 

Reviewer 3 Report

The authors have done a great job at dealing with sky and ground segmentation problem. The stated problem is important and studied by many authors. Proposed approach seems to be better comparing with the state of the art methods.

suggestions:

  1. Authors reported that designed method can meet real-time requirements compared with related work. Please provide provide proof of improvement especially timing, AUC curves, etc
  2. Please include time analysis. 
  3. Compare performance with simple classic methods like Otsu based segmentation.
  4. The paper can't be published in such large volume! 31 pages will be not be appreciated by mdpi readers, please respect readers time and make it easy to read. Article is not scientific report!  

Round 2

Reviewer 3 Report

Thank you. This version can be accepted.

This manuscript is a resubmission of an earlier submission. The following is a list of the peer review reports and author responses from that submission.

Round 1

Reviewer 1 Report

This paper proposes a method to deal with the segmentation into sky and ground on claimed navigation visions of planetary rovers. The approach is based on deep learning.

I make a differentiated evaluation of this manuscript. On one side, the methodological approach seems valid to segment sky from ground on navigation landscape imagery with very good performances (a bit better than the alternatives), but on another the focus and application of the approach is not on the planetary imagery as indicated. This is misleading, since the title, the abstract and the way the review of the state of the art are presented always point to planetary rover visions. Although it may be very useful for planetary rover navigation, I consider this mention now to be a bait only since it is not seriously focused on that real and specific problem.

I was very deceived when the datasets are introduced, all related to terrestrial examples. Even the use of a dataset with analogue terrestrial imagery and the transfer of the trained model to half a dozen Martian images are also not enough nor convincing. Although the outputs are visually nice you can’t state based on six images that ‘This research… successfully transfers the pre-trained state-of-the-art segmentation model to the planetary rover vision.’ You know you can’t.

In addition, what is the utility (or advantage) of training a classifier with samples that include features only available in terrestrial views (buildings and other human constructions, trees and vegetation, water, very cloudy skies, etc) and then transfer the trained model to images on other planetary grounds without those features? I admit that terrestrial views may be more complex and varied than the extra-terrestrial ones, but this is not a good justification.  

Also, how do you guarantee that the specific Earth’s atmospheric characteristics are transferable to other planetary bodies with very distinctive atmospheres (also resulting in differentiated types of illuminations) like Mars, Venus or Titan or even without an atmosphere (like the Moon)?

Although I agree with the justification that ‘To transfer the proposed framework to the onboard device requires a few more steps’, I see it as a bad and inadequate excuse in the context of the current manuscript. It doesn’t justify why the proposed framework wasn’t fully tested in real images captured by planetary rovers outside the Earth. There are plenty of these images captured in-situ by rovers and landers, as mentioned by the authors in the introduction, that must have been used instead.

It means that this contribution as it is now, is not suitable for publication. But since I think the method is valid and adequate to deal with the problem in hands, I suggest the authors to choose one of two ways to solve the ambiguity shown.

One, by presenting the application to be more generic and applied in different types of images and where the rover visions on analogue terrestrial datasets are only one among many other applications. The other, by really focusing on planetary datasets with images captured in-situ on Mars (but also on other planetary bodies if desired like, for instance, the Moon). The first option is simpler, mainly to refocus the aims and to reorganize and rewrite some parts of the paper, but much less appealing in my opinion. The second one needs additional efforts and developments as it will require building your own annotated dataset or datasets with adequate dimension and diversity of scenes before testing the novel approach and the concurrent ones. It is up to the authors to decide, but I consider the second option much more original and important for planetary navigation, which would expectedly have a much higher relevancy and enduring impact.

Reviewer 2 Report

This work presents a sky and ground segmentation framework based on weak supervision and transfer-learning for the navigation visions of the planetary rover. The employed hardware configuration is i7-7700 CPU, RTX1080 graphics card, 32GB memory. Results verifies better performance for all seven metrics compared to state-of-the-art.

Some issues should be addressed before this manuscript could be considered for publication.

1 Disadvantages and limitation of the proposed framework should be mentioned.

2 Regarding real-time implementation, a discussion is required to address the feasibility of transferring the proposed framework to the onboard device.